# Ecological Study on Thyroid Cancer Incidence and Mortality in Association with European Union Member States’ Air Pollution

**DOI:** 10.3390/ijerph18010153

**Published:** 2020-12-28

**Authors:** Evanthia Giannoula, Christos Melidis, Savvas Frangos, Nikitas Papadopoulos, Georgia Koutsouki, Ioannis Iakovou

**Affiliations:** 1Second Academic Nuclear Medicine Department, Academic General Hospital of Thessaloniki “AHEPA”, Aristotle University of Thessaloniki, Kiriakidi 1 St, 546 21 Thessaloniki, Greece; eva_giann@hotmail.com; 2Radiation Therapy Department, CAP Santé, 13 Rue Marcel Paul, 20200 Bastia, France; melichristos@hotmail.com; 3Department of Nuclear Medicine, Bank of Cyprus Oncology Centre, 32 Acropoleos Avenue, Strovolos, Nicosia 2006, Cyprus; savvas.frangos@gmail.com; 4General Hospital of Thessaloniki “Georgios Gennimatas”, Ethnikis Aminis 41 St, 546 35 Thessaloniki, Greece; doctornikitas@yahoo.gr; 5Medical School, Aristotle University of Thessaloniki, 4124 Thessaloniki, Greece; koutsoukig@gmail.com

**Keywords:** thyroid cancer, air pollution, ecological study

## Abstract

Over the last few decades, thyroid cancer incidence has had a significant increase. Despite well-known genetic and epigenetic factors (radiation, overdiagnosis, already existing benign thyroid tumors), the effect of air pollution on its incidence and mortality has not yet been fully elucidated. In this study, air pollution data from 27 EU member states is used in order to analyze its association with thyroid cancer incidence, and mortality and socioeconomic factors are examined as confounders. This ecological study used age standardized thyroid cancer incidence and mortality rates per 100,000 people for the year 2012 from 27 EU member states, collected from the International Agency for Research on Cancer, World Health Organization and European Cancer. Data regarding mean air pollutant mass concentrations for 1992, 2002 and 2012 was collected from the European Environment Agency. Data analysis was carried out using Prism 5.0 and SPSS v.20. Multiple regression analysis showed a statistically significant positive association between thyroid cancer incidence in men and the environmental 2012 masse of Benzo (k) Fluoranthene (r^2^ = 0.2142, *p* = 0.042) and HexaChlorocycloHexane (r^2^ = 0.9993, *p* = 0.0166). Additionally, a statistically significant positive association was observed between the thyroid cancer mortality rate in men and the 1992 environmental concentrations of Hg (r^2^ = 0.1704, *p* = 0.043). Data indicates that some air pollutants may have an effect on increased thyroid cancer incidence and mortality, at least in men. However, causal relationships cannot be fully supported via ecological studies, and this article only focuses on the EU and uses only three distinct time periods.

## 1. Introduction

Thyroid cancer is the most common type of endocrine cancer worldwide with two thirds of cases diagnosed in people under 55 years old. During recent decades, it has the fastest growing incidence among all types of cancer for most countries, while developed countries experience this phenomenon twice as fast for both men (2.2/100,000 and 1/100,000 respectively) and women (5.5/100,000 and 2.6/100,000 respectively) [1]. In contrast with other types of cancer, including breast, colon, lung and prostate, where mortality rate has decreased over the last two decades, thyroid cancer mortality remains rather stable (yet lower compared to other malignant neoplasms—0.5 cases per 100,000 individuals), presenting a slight increase of +0.8% per year [2]. This increase, which is mainly true for men, is observed despite earlier diagnosis and more appropriate treatment being offered to patients. Confirmed factors associated with incidence and/or mortality increase of thyroid cancer are overdiagnosis [3,4,5,6,7], ionizing radiation exposure [8,9,10], genetic predisposition [11,12], benign thyroid diseases [13,14], hormonal and reproductive factors [15] and diet [16]. Thus, disease rates are expected to be higher in populations with an increased standard of living and socio-economic statue, also due to better access to health care [6].

Air pollution levels in the European Union (EU) have declined since the late 20th century, but they are still quite high. The World Health Organization (WHO) and the International Agency for Research on Cancer (IARC) have identified air pollution as potential carcinogen for humans [17], and extensive epidemiological and experimental studies have linked the concentrations of specific pollutants in the atmosphere to the occurrence of certain types of malignancies [18], mainly lung cancer [19]. However, to our knowledge, no environmental factor has been studied yet for thyroid cancer within the EU countries.

Ecological studies are observational studies carrying findings that concern groups or the entire population in question. In this context, the aim of this ecological study is to explore possible effects of air pollution on pathogenesis or burden of malignant neoplasms of the thyroid gland on a pollutant-by-pollutant basis. This will be carried out using incidence and mortality data of 2012 for EU member states, and since exposure to air pollution carries a rather difficult to estimate latency time, air pollution data dating 20, 10 and 0 years before 2012 will be used. In order to limit the possibility of ecological fallacy, ensure reliability of the results and reduce confounding effects, the impact of socio-economic level of the member states under investigation will also be checked for the years 1992, 2002 and 2012.

## 2. Materials and Methods 

Data on thyroid cancer incidence and mortality for 2012 were derived from the IORC website [20] for the 27 member states of EU (Croatia only joined the EU in 2013). The European Environment Agency for Air Pollution website was accessed on 30/06/2014 [21,22] with the following indication: “National emissions reported to the Convention on Long—range Transboundary Air Pollution (LRTAP Convention)”, so that the air pollution data of the 27 member states is collected for years 1992, 2002 and 2012.

The World Bank Data Group [23] of the World Bank Group was used as a source of socioeconomic assessment data for the same member states and for the same three years, with the following fields: “World Development Indicators”, “Education Statistics—All Indicators”, “Gender Statistics ” and “Health Nutrition and Population Statistics”. For some indicators there was overlap for the three categories and attempts were made from each category to select indicators for which data were available for all member states during the three years under review, avoiding repetitions and overlaps. Special care was taken so that the selected indicators would make any differences between sex, age group, urban and rural population noticeable. 

Statistical dependence analysis had age-adjusted indicators of incidence and mortality as dependent variables and the independent variables were air pollutant masses and socio-economic indicators, but not the smoking habits of the individuals, since this data was not available. The following analyses were performed using GraphPad Prism v.5 software, for the corresponding indicators in men, women and both sexes: Kolmogorov-Smirnov Test (KS-test), in order to test the regularity of the distribution of incidence and mortality rates and single-factor linear regression, concerning the association of incidence and mortality rate with air pollution levels and, independently, the level of socio-economic indicators of the years 1992, 2002 and 2012. In cases in which statistically significant association with air pollution levels as determinants were observed, the IBM Statistical Package for Social Sciences (SPSS) v.20 (IBM Corp., Armonk, NY, USA) was used for multi-factor linear regression in order to test the association with the level of socio-economic indicators (as confounding factors). A statistically significant level of 5% (*p* < 0.05) was used in all analyses. 

## 3. Results

A normal distribution was found for thyroid cancer incidence for both sexes as a whole (*p* = 0.099), as well as for the mortality rate for women (*p* = 0.058). For the remaining epidemiological indicators tested, no normal distribution was found (Table 1).

Since a normal distribution was observed in two out of six dependent variables, in order to perform a statistical dependency test, values of the other four indicators of Table 1 were turned into logarithms (ln).

Among the EU member states that are present in the first (worst) three countries carrying the highest number of air pollutants at maximum levels are France (all years of this study), Poland (all years of this study), the United Kingdom (years 1992 and 2002) and Germany (years 2002 and 2012), as can be seen at Table 2.

Single-factor linear regression (no correction for socio-economic factors) analysis for 1992, 2002 and 2012 for men, women and both sexes for the association of data from Table 1 and Table 2 give the following statistically significant results, also presented in Table 3:Male thyroid cancer incidence with benzo (k) year 2012 levels (*p* = 0.046).Male mortality rate with Hg year 1992 levels (*p* = 0.04).Male mortality rate with HCH year 2012 levels (*p* = 0.017).

As can be seen in Table 3, male mortality associated with HCH levels in 2012 is a result of only three records. Therefore, as it does not constitute a real dependence, it is not taken into account. Thus, single-factor linear regression analysis was performed for the association of the first two results of Table 3 with the levels of socio-economic indicators for the three years in question. Results that are statistically significant are presented in Table 4.

Multiple regression analysis was performed for all findings of Table 4. The results of which are presented in Table 5.

No association was observed (Table 5), thus socio-economic indicators cannot be considered as confounding factors.

Summary statistics for benzo (k) and Hg results presented in the above tables are shown in Table 6.

## 4. Discussion

Significant progress has been made in understanding the biology and molecular pathways of thyroid carcinogenesis. However, far less progress has been made in identifying a clearly stated and concretely documented risk profile for thyroid cancer [5]. Epidemiological and experimental studies have linked the concentrations of specific pollutants in the atmosphere to the occurrence of certain types of malignancies [18,24]. Air pollution is a major public health problem linked with several diseases. Economic growth, urbanization, industrialization, energy consumption, transportation and population growth are factors that exacerbate the problem, especially in large EU cities [25]. Therefore, it is not surprising that the results of this ecological study, based on the EEA census report for the years 1992–2012, show that the highest levels in most of the gas pollutants examined were found in the largest and most industrialized countries in the EU. Additionally, a high standard of living is positively associated with increased epidemiological rates of thyroid cancer [26]; however, there is no confounding effect of socioeconomic level on the relationship between disease and air pollution.

Benzo (k) or benzo (k) fluoranthene is an organic compound with the chemical formula C20H12 and belongs to the polycyclic aromatic hydrocarbons (PAH). PAHs are metabolized by cytochromes P4501A1 and P4503A4 to exhibit their toxic effects and are associated with an increased risk of lung cancer and skin cancer through mutations in the p53 tumor suppressor gene [18]. Although exposure to elevated PAH levels has long been blamed for cancer of the urogenital system and malignancies of the gastrointestinal tract, larynx and pharynx, the literature review did not reveal any data linking PAHs to a specific mechanism with thyroid cancer or to the development of other thyroid conditions that could potentially lead to the development of malignancy. Therefore, a possible mechanism for the development of thyroid cancer after exposure to benzo (k) is the mutation in the p53 tumor suppressor gene. In fact, the latency time, which in this case corresponds to 20 years, may reinforce the original hypothesis.

Mercury may directly affect thyroid function by interrupting the synthesis and/or secretion of thyroid hormones, but its main toxic effect is indirect and concerns disorders of the hypothalamic-pituitary-thyroid axis. Specifically, the inhibition of 5-deiodionase or induction of hepatic microsomal enzymes such as T4-uridine bisphosphonate glucuronyltransferases are described as possible mechanisms. Through these mechanisms, it is possible to reduce the levels of circulating T4 and T3 and lead to increased secretion of thyroid stimulating hormone (TSH) from the pituitary gland. The chronic overexpression of TSH predisposes the thyroid gland to the development of hyperplasia and neoplastic lesions through epigenetic mechanisms associated with hormonal disorders [27]. The study of the mechanisms of induction of mutations and carcinogenesis as a result of exposure to heavy metals is the subject of systems toxicology. Systemic toxicology may be able to provide answers to the type of changes that mercury exposure to the human genome causes in order to qualitatively explain the quantitatively observed increase in male rates of mortality from thyroid cancer in the EU found in the present study [28].

Ecological studies carry results particularly prone to the effect of confounding factors and errors that cannot always be calculated at the population level. In addition, since they concern a certain population-region, ecological fallacy occurs when the conclusions drawn for the study population are arbitrarily considered to be valid at the individual level and special attention is required when extending their conclusions to a wider range of studied populations-regions. In general, ecological studies are considered reliable if they meet the necessary specifications, especially when examining causal associations of epidemiological indicators with exposure to risk factors for the occurrence of diseases in the population or when referring to a population in this socio-economic context [29]. Additionally, thyroid cancer is of a multifactorial nature, and, thus, it is necessary to control other factors that may affect the association. Lastly, our findings are limited by the fact that air pollution data from only three years were used and thyroid cancer has a non-negligible latency time, thus the action of benzol(k) from 2012 on humans in the same year is doubtful.

## 5. Conclusions

Our results are an indication that benzo (k) levels are correlated with thyroid cancer incidence and Hg levels with mortality, both in males, and do not confirm that socio-economic indications are confounding factors. The effect of air pollution on malignant neoplasms of the thyroid is a field that needs further study on the pathogenesis of the disease, so that preventive measures can be taken and better control of the disease is achieved.

## Figures and Tables

**Table 1 ijerph-18-00153-t001:** Kolmogorov-Smirnov Test (KS-test) testing the regularity of incidence and mortality distribution for 2012 for 27 EU member states, age-adjusted for 100,000 people.

	Incidence (Both Sexes)	Mortality (Both Sexes)	Incidence (Men)	Mortality (Men)	Incidence (Women)	Mortality (Women)
Minimum	1.9	0.1	0.7	0	3.1	0
25% Percentile	3.6	0.4	1.9	0.3	5.4	0.4
Median	5.3	0.5	2.9	0.4	7.6	0.5
75% Percentile	8.9	0.5	3.8	0.5	13.8	0.6
Maximum	15.5	0.7	8	0.7	24.2	0.9
Average	6.519	0.47	3.13	0.418	9.704	0.485
Standard deviation SD	3.574	0.141	1.739	0.136	5.421	0.175
Typical mid-range errors	0.688	0.027	0.335	0.026	1.043	0.034
*p* value	0.099	<0.001	0.036	0.017	0.045	0.058
Normal distribution (alpha = 0.05)	Yes	No	No	No	No	Yes

**Table 2 ijerph-18-00153-t002:** Air pollution data of 27 EU countries for 1992, 2002 and 2012.

Year	Classification	Country	Number of Air Pollutants at Maximum Levels	Type of Air Pollutants at Maximum Rates
1992	1st	United Kingdom	10	benzob, Hg, NMVOC, NOx, Pb, HCH, HCB, PCB, Se, SOx
	2nd	France	6	CO, Cr, NH_3_, PM 10, PM 2.5, TSP
	3rd	Poland	5	benzo k, Cd, Indeno, Ni, Zn
2002	1st	France	6	CO, NH_3_, NMVOC, PM 10, PM 2.5, TSP
	1st	Poland	6	benzo a, benzo b, benzo k, Cd, Indeno, Pb
	2nd	Germany	5	Cr, Cu, Hg, NOx, Zn
	2nd	Spain	5	Ni, Se, SOx, HCB, PAH
	3rd	United Kingdom	2	HCB, PCB
2012	1st	Poland	9	benzo a, benzo b, Dioxins and Furans, Cd, Indeno, Ni, Pb, PCB, SOx
	2nd	Germany	6	CO, Cr, Cu, NMVOC, NOx, Zn
	3rd	France	4	NH_3_, PM 10, PM 2.5, TSP

**Table 3 ijerph-18-00153-t003:** Thyroid cancer incidence and mortality associations with air pollutants.

Epidemiological Index (Age-Modified per 100,000 Population)	Sex	Air Pollutant, Year	*p*-Value	Number of Records ^†^
Incidence	Men	Benzo (k), 2012	0.046	19
Mortality	Men	Hg, 1992	0.04	25
Mortality	Men	HCH, 2012	0.017	3

†: Although the total number of recordings for each air pollutant should be equal to the number of EU member states (27), there were cases where there was no data for some countries.

**Table 4 ijerph-18-00153-t004:** Statistically significant results of single-factor linear regression analysis between incidence and mortality in men with socio-economic factors.

Epidemiological Index (Age-Modified per 100,000 Population)	Socio—Economic Index	*p*-Value	Number of Records ^†^
**Incidence in men, Benzo (k) 2012 levels**	Gross National Income (GNI) per capita, purchasing power parity (PPP) (current international $)	0.043	27
**Mortality in men, Hg 1992 levels**	Adjusted net national income per capita (constant 2005 US$)	0.040	22
	Adjusted net national income per capita (currentUS$)	0.010	24
	Hospital beds (per 1000 persons)	0.036	24
	Life expectancy at birth, total (years)	0.048	26
	Labor force participation rate, male (% of male population, ages 15–64)	0.034	26
	Gross Domestic Product (GDP) (current US$)	0.013	24
	GNI per capita, PPP (current international $)	0.040	22
	Gross enrolment ratio, all levels combined (except pre-primary)for both sexes	0.007	23

†: Although the total number of records for each socio-economic factor should be equal to the number of EU member states (27). There were cases where there was no data for some countries.

**Table 5 ijerph-18-00153-t005:** Multiple regression analysis of incidence and mortality associated with Benzo (k) 2012 and Hg 1992 levels with statistically significant socio-economic factors.

		Unstandardized Coefficients		*p*-Value
		B	Std. Error	
Incidence in men. Benzo (k) 2012 levels	Emissions—Gg (10^6^ Kg)	−30.477	17.860	0.107
	Gross National Income (GNI) per capita, purchasing power parity (PPP) (current international $)	0.003	0	0.753
Mortality in men, Hg 1992 levels	Emissions—Gg (10^6^ Kg)	−3.473	4.501	0.466
	Gross enrolment ratio. All levels combined (except pre-primary). Total	−0.002	0.009	0.804
	Adjusted net national income per capita (constant 2005 US$)	−0.028	0	0.495
	Adjusted net national income per capita (current US$)	0.072	0	0.478
	Hospital beds (per 1000 people)	−0.015	0.028	0.62
	Life expectancy at birth, total (years)	−0.095	0.054	0.125
	Labor force participation rate, total (% of total population ages 15–64)	−0.040	0.019	0.078
	Gross Domestic Product (GDP) (current US$)	−0.028	0	0.706
	GNI per capita, PPP (current international $)	0.031	0	0.544

**Table 6 ijerph-18-00153-t006:** Descriptive statistics for benzo (k) and Hg-Gg (10^6^ Kg).

	Emissions—Benzo (k)—Gg (10^6^ Kg)	Emissions—Hg—Gg (10^6^ Kg)
Mean (sd)	0.004 (0.006)	0.008 (0.01)
Median (IQR ^a^)	0.001 (0.0001, 0.004)	0.003 (0.001, 0.011)
Minimum	0	0
Maximum	0.027	0.04

^a^ IQR: Interquantile Range.

## Data Availability

The data presented in this study are openly available in https://ecis.jrc.ec.europa.eu/?Cancer=35, https://www.eea.europa.eu/data-and-maps/data#c11=&c17=&c5=all&c0=5&c_start=0, http://www.eea.europa.eu/data-and-maps/data%23c11=air&c17=&c5=all&c0=5&b_start=0 and https://www.eea.europa.eu/data-and-maps/data/national-emissions-reported-to-the-convention-on-long-range-transboundary-air-pollution-lrtap-convention-8.

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
