# Peer review of "Ecological Study on Thyroid Cancer Incidence and Mortality in Association with European Union Member States’ Air Pollution"

_ijerph, 2020, doi:10.3390/ijerph18010153_

Round 1
Reviewer 1 Report
The interesting manuscript “Ecological study on thyroid cancer incidence and mortality in association with European Union member states’ air pollution” analyze the relationship between air pollution and thyroid cancer incidence and mortality in EU. Since smoking is known to affect thyroid function, this parameter should also be considered in the analysis. The manuscript is well written and within the scope of the journal.
Results
Table 1: Authors might correct the use of comma and period in the values. Moreover, authors should include the unit for incidence and mortality.
Table 2: Authors might include a legend with the definitions of the abbreviation used.
Diagram 1: The graphs should be identified with letters and the legend should include a description of each one. In the second and third graphs, why was the mortality expressed as negative value? Why did first and second graphs have many values while the third have only 3?
Table 3: Both diagram 1 and table 3 describe the same result. Authors might choose one and exclude the other.
Tables 4 and 5: Authors might include a legend with the definitions of the abbreviation used.
Graphs showing thyroid cancer incidence and mortality in male along the years (1992, 2002 and 2012) besides graphs showing Benzo(K) and Hg levels along the years could be of interest. The exposition to pollutants could lead to long-term consequences, so maybe high levels of pollutants in a moment could be related to high thyroid cancer incidence ten years late, for example. Therefore, time-course data could be useful to this inference.
Reviewer 2 Report
In this ecological study, the authors investigated an association between thyroid cancer incidence and mortality and some specific air pollutants in European Union member states.
Major comments
Comment 1: Throughout the text, the authors stated that they used linear regression technique to test correlation between dependent and independent variables. Taking into account that the authors performed linear regression analysis it is better to use term “to test an association” instead of “to test correlation” as they didn’t calculate and provide correlation coefficients.
Comment 2: What did authors mean by “confusing “ factors? Is it confounding factor? If it is, please use term confounding factor or confounder. Also, please define criteria used to make conclusion whether the variable is confounder or not in multivariable regression analysis. Usually variable is stated as confounder if it’s inclusion in the model changes coefficient of main effect by at least 10% .
Comment 3: Line 100: Is this age-adjusted incidence and mortality rate per 100,000? Please write the full name of the indicator.
Comment 4: Line 103: Did you obtain normal distribution of indicators after logarithmic transformation as a condition to apply linear regression? Usually, Poisson regression is performed in analysis of association between incidence/mortality rates and effect of exposure to potential carcinogen. The reason to use Poisson regression is the fact that incidence/mortality rates are count variables by its nature (not continuous) so they can’t have negative value in contrast to the continuous variable that is defined in the interval -∞, +∞. It would be good to test whether indicators have Poisson regression, to perform Poisson regression analysis and to compare with present results.
Comment 5: It is not completely clear what are standardized and unstandardized coefficients in Table 5. Did the authors mean adjusted and unadjusted coefficients or something else? Do coefficients describe benzo(k) levels or added socioeconomic parameter? Is benzo(k) level main effect in analysis adjusted for socioeconomic parameter? If so, it would be good to see coefficients of main effect before and after adjustment to give proper conclusion whether the adjusting variable is confounder or not. (see comment 2).
Comment 6: On the Diagram 1, did the authors transform variables describing the benzo(k) and HCH measurement levels because there are negative values on the y axis?
Some minor comments:
Comment 6: Please change Gg to 106 kg in the label of y axsis on Diagram 1.
Comment 7: Please provide some summery statistics (mean, median, etc.) for benzo(K) and Hg measurements for which results in tables were presented.
Reviewer 3 Report
Giannoula and co-workers is an observational study on thyroid cancer incidence and mortality in association with European Union member states’ air pollution based on the analysis of data collected from the International Agency for Research on Cancer, World Health Organization and European Cancer for the year 2012 from 27 EU member states. To the author’s knowledge, at the time of this study, no environmental factor has been studied yet for thyroid cancer. However, work recently published in the Int J Environ Res Public Health named Role of Emerging Environmental Risk Factors in Thyroid Cancer: A Brief Review, includes references to previous work associated to environmental pollutants and thyroid conditions and thyroid cancer risk associated with environmental factors. Also, published in 2018 an article entitled Air pollution from industrial waste gas emissions is associated with cancer incidences in Shanghai, China at Environ Sci Pollut Res Int. that includes thyroid cancer association with air pollutants. I suggest authors specify work originality by changing line 59-60. I understand data analyzed source refers specifically to 27 EU member states and also regarding air pollution and Thyroid cancer incidence for the year 2012 and for the reason that exposure to air pollution consequences regarding time are difficult to be estimated t, air pollution data from 1992 and 2002 were included on this study.
Diagram 1 is hard to interpret by itself. Legend to graph could be more explanatory.
Considering that a thyroid cancer risk factor is anything that increases the probability of someone to develop thyroid cancer, other risk factors are known and well documented including gender, age, diet, inherited conditions, genetic mutations, and others. I believe the authors meant to say documented risk caused by exposure (line 137).
Reviewer 4 Report
Giannoula et al. performed research which was focused on the evaluation of a potential relationship between thyroid cancer incidence and mortality and air pollution in 27 EU members. The topic of present manuscript I found as quite interesting, because the process of identifying causes of dramatically increase of thyroid cancer incidence over the last 2-3 decades is still open and hot problem. However, I am not sure whether this study might be clinically useful. Nevertheless, some readers can find it as thought-provoking, especially clinicians in France, Poland, Germany or the United Kingdom, which are reported in the study as countries of the most air polluted in all of three analysed time points.
I suggest an additional reviewing a manuscript by an authority in a field of medical statistic methodology.
Discussion is well-balanced, a limitation due to ecological and population-based design of the study is noticed by Authors. Usefulness of the present study results at an individual level in particular patients living in polluted regions remains unclear.
Several comments are listed below:
- All abbreviations used in the text (e.g. line 115: HCH) should be expanded when they are described at first. To improve clear reading the study results, I suggest listing and explaining all the analyzed air pollution factors (CO, PCB, SOx, etc.) in the main text. On the other hand, all abbreviations (e.g. NMVOC, GNI, TSP, Gg, etc.) used in tables also should be clearly explained under the tables.
- Authors could explain the statistical role of a parameter ‘r2’ used in results section, as they described a statistical significance of p-value.
- Table 1. What does ‘.e’ mean at the end of ‘typical mid-range error.e’? and ‘?’ in the last row of this table?
- Table 4 is poorly readable.
Round 2
Reviewer 2 Report
The authors improved the quality of the manuscript. There is only one additional comment related to the use of decimal points and commas in writing the numbers.
The paper was written in English language so English language grammatical rules must be applied. Please correct the use of commas and decimal point consistently throughout the text. Note that in UK/US English a decimal point, and not a comma, is placed as separator before the cents (the fractional part of the decimal number). On the other side, in US/UK English a comma, and not a point, is placed as a 3-digit group separator.
As I mentioned, the paper is written in English language and people from different parts of the world will read it without previous knowledge on how the numbers are written in other languages.
Author Response
Thank you once more for your review. You are absolutely correct and we are sorry for the inconvenience (we actually also found that half of our numbers used "commas" and the other half "points", although they were all decimal numbers). Now this is corrected, as per your proposal.